# Assessment of Genetic Damage Induced via Glyphosate and Three Commercial Formulations with Adjuvants in Human Blood Cells

**DOI:** 10.3390/ijms24054560

**Published:** 2023-02-25

**Authors:** Carlos Alvarez-Moya, Mónica Reynoso-Silva

**Affiliations:** Environmental Mutagenesis Laboratory, Cellular and Molecular Department, University of Guadalajara, Guadalajara 45200, Jalisco, Mexico

**Keywords:** glyphosate, herbicides, adjuvants, genetic damage, genotoxicity, migration groups

## Abstract

There is considerable controversy regarding the genotoxicity of glyphosate (N-(phosphonomethyl) glycine). It has been suggested that the genotoxicity of this herbicide is increased by the adjuvants added to commercial formulations based on glyphosate. The effect of various concentrations of glyphosate and three commercial glyphosate-based herbicides (GBH) on human lymphocytes was evaluated. Human blood cells were exposed to glyphosates of 0.1, 1, 10 and 50 mM as well as to equivalent concentrations of glyphosate on commercial formulations. Genetic damage (*p* < 0.05) was observed in all concentrations with glyphosate and with FAENA and TACKLE formulations. These two commercial formulations showed genotoxicity that was concentration-dependent but in a higher proportion compared to pure glyphosate only. Higher glyphosate concentrations increased the frequency and range of tail lengths of some migration groups, and the same was observed for FAENA and TACKLE, while in CENTELLA the migration range decreased but the frequency of migration groups increased. We show that pure glyphosate and commercial GBH (FAENA, TACKLE and CENTELLA) gave signals of genotoxicity in human blood samples in the comet assay. The genotoxicity increased in the formulations, indicating genotoxic activity also in the added adjuvants present in these products. The use of the MG parameter allowed us to detect a certain type of genetic damage associated with different formulations.

## 1. Introduction

Glyphosate (N-(phosphonomethyl) glycine, C_3_H_8_NO_5_P) (G) is the most popular herbicide in the world. It is distributed as commercial glyphosate-based herbicide (GBH) formulations [1,2,3], which have compounds (adjuvants) that improve penetration through the leaves; therefore, G is not used in its pure form [4,5]. Human exposure to CFG occurs during occupational and environmental activities and a maximum urinary glyphosate concentration of 1.7 times above the recommended acceptable daily intake has been reported [6]. There is abundant evidence of G genotoxicity [7,8,9,10] and its effect on the expression of some genes and in human health [11,12]. Additionally, G represents a risk for other living organisms [13]. However, the absence of its genotoxic activity is also reported [14,15]. Several studies suggest that it is carcinogenic [11,16,17] and contributes to the development of non-Hodgkin lymphoma in humans [18], however, there is controversy among the IARC, regulatory authorities, and scientists regarding its carcinogenic potential [3,16,18,19].

Some research suggests that the adjuvants present in GBH are responsible for the genotoxic activity and not G [1,20,21]; however, this is also highly controversial because other studies report the opposite [14,15,21,22,23,24,25]. Due it is considered confidential business information (CBI), there is no information available about the adjuvants and their concentration in each GBH, and their genotoxic properties are unknown. Some known adjuvants are sodium dodecyl benzene sulfonate, sodium alkyl polyglucoside, lauryl glucoside, and polyethoxylated adjuvants [5,20]. For this reason, a reassessment of the GBH safety standards is suggested [26]. There is evidence that the genotoxic effect of G and GBH can occur in two ways: as a direct effect on DNA [27] or by the induction of oxidative stress [23]; obviously more information is required in this sense.

A wide variety of test systems exist to assess genotoxicity; however, the comet test is a rapid bioassay and detects different types of DNA damage [28,29,30] and is particularly used with human lymphocytes [8]. Using the comet test, Reynoso-Silva et al. [31] reported a new method: migration groups (MG) for the detection of genetic damage related to different sensitivities in cells in response to specific genotoxic agents. Blood cells offer advantages for monitoring the effects of genotoxic agents [32] and additionally, most of the published works were focused on the in vitro effects on human lymphocytes of high and low glyphosate concentrations [33].

Although the genotoxicity of G is highly controversial, the growing evidence of its genotoxic activity makes it necessary to also evaluate the role of the adjuvants present in GBH as enhancers of G genotoxicity or as additional genotoxic agents. Commonly, these adjuvants are not identified on the commercial labels, alluding to industrial property rights. In this work, the genotoxic activity was evaluated in human lymphocytes of different G concentrations and of three GBH. Three different evaluation parameters were also used, including: tail length, tail moment, and migration groups, with the latter recently proposed. For G and GBH previously reported concentrations were used.

## 2. Results

### 2.1. Genetic Damage Induced via G and GBH Using Tail Length and Tail Moment

The genotoxic activity of various concentrations of G and GBH (FAENA, TACLE and CENTELLA) using the tail length parameter is shown in Figure 1. All concentrations of pure G induced significant genetic damage (*p* < 0.05) compared to NC and even higher than PC (*p* < 0.05) with the exception of the 0.1 mM concentration. Concentrations of G 1, 10 and 50 mM showed practically the same level of genetic damage. All FAENA concentrations (1–100 mM) induced significant genetic damage (*p* < 0.05) compared to NC and dose-dependent concentrations, although 0.1- and 1-mM concentrations showed the same level of damage. The magnitude of damage was similar to G. TACKLE induced dose-proportional genetic damage and to a greater extent than that induced via G and FAENA. CENTELLA showed significant genotoxic activity (*p* < 0.05) with respect to NC only at the 100 mM concentration. The magnitude of genetic damage induced via the highest concentrations of G (50 mM), with respect to FAENA (100 mM) and TACKLE (100 mM), was significantly lower (*p* < 0.05). Similarly, differences were observed between FAENA 100 mM and TACKLE 100 mM. On the other hand, CENTELLA 100 mM showed significantly lower genetic damage than that induced via FAENA 100 mM and TACKLE 100 mM (Figure 1).

The evaluation of the genotoxic activity by means of the tail moment parameter of the aforementioned compounds is presented in Figure 2. Only the CENTELLA 100 mM concentration showed genotoxic activity; all the concentrations of FAENA, TACKLE, and G (with the exception of 0.1 mM) did not. The 100 mM FAENA and 100 mM TACKLE concentrations were significantly different from 50 mM G. A significant difference was also observed between FAENA 100 mM and TACKLE 100 mM. 100 mM CENTELLA showed a significant difference with 100 mM FAENA and 100 mM TACKLE. With the exception of CENTELLA 100 mM, the behavior of genotoxicity through tail moment of the compounds studied was practically the same as that described for the tail length parameter.

### 2.2. Genetic Damage Induced via G and GBH Using Migration Groups

The MG parameter allowed for the evaluation of the genetic damage induced via glyphosate and GBH. Comets with the same amount of damage were identified and grouped (tail migration range indicated in µm) and the most frequent migration group was indicated (Figure 3). The NC showed MG with little migration and the most frequent group reached 40% (2–4 μm).

As observed in Figure 3, all concentrations of PC, G, and GBH increased the percentage of the most frequent migration group. The increase in the concentration of G also broadened the migration range of the cauda of the most frequent MG: G 0.1 mM (2–6 µm) (57.7%), G 1 mM (4–8 µm) (53.3%), G 10 mM (5–10 µm) (72.6%), and G 50 mm (6–10 µm) (72.2%). FAENA and TACKLE showed similar behavior with the exception of 0.1 mM; in this case, the migration ranges are higher than those produced by the 1- and 10-mm concentrations. As the NC, CENTELLA showed the same migration ranges for all concentrations (2–6 µm); however, these groups presented frequency percentages close to 90%. CENTELLA 100 mM produced a slight increase in the migration range (3–6 µm), but the frequency of this group decreased to 51%.

## 3. Discussion

Although there is evidence of the genotoxic activity of G [7,8], this is controversial [16,18], and it is necessary to increase the degree of certainty of its genotoxicity. Alvarez-Moya et al. [34] reported the genotoxicity of G in the blood of humans in a mean concentration of 0.0007–0.7 mM. To ensure the analysis of genotoxic activity, high concentrations (50 and 10 mM) and lower ones such as those reported by Khan et al. [7] were used, observing the dose-dependent genotoxic activity of G, from 0.1 mM–50 mM, using the tail length, tail moment, and MG parameters.

Some reports suggest that the genotoxic activity of GBH is due to the adjuvants used and not to G [1,20]. Our data clearly showed the genotoxicity of G and GBH. After calculating the concentration of G and subsequent evaluation of FAENA and TACKLE, it was observed that these induced genetic damage at the same concentrations with which the genotoxicity of G was evaluated, effectively indicating the presence of G as a genotoxic agent; however, the FAENA 100 mM concentration and all concentrations of TACKLE showed genotoxic activity greater than that of G and dose-dependent concentrations, which can be attributed to the additional genotoxic effect of adjuvant substances as reported by Nagy et al. [1] or to possible glyphosate-adjuvant synergism, as has been observed in other types of glyphosate interactions [35,36].

Only the 100 mM CENTELLA concentration induced genetic damage and an antagonistic response, possibly due to the use of adjuvants different from those found in FAENA and TACKLE, which prevented the genotoxic activity of G at the lowest concentrations. In this way, the differences in the genotoxic activity of the GBH studied indicate the use of different adjuvants. Because the adjuvants used are not identified on the label (industrial law), it is difficult to specifically identify genotoxic agents.

This factor and others such as the use of different test systems [37] fuel the controversy surrounding the genotoxicity of G [16,18]. Another possibility is the lack of veracity about the actual G content: the GBH use sheet indicates the glyphosate content—FAENA (363 g/L), TACKLE (360 g/L), and CENTELLA (360 g/L) (see additional material)—and allowed us to prepare dilutions of up to 100 mM or higher. We consider that these data may be inconsistent, since the maximum concentration of G obtained in our laboratory (using water as diluent and at room temperature, as indicated on the labels) was 50 mM.

The comet test is a very useful, fast, and efficient tool to assess DNA damage [38,39,40,41]. Various parameters are used to assess genetic damage, such as the tail length and tail moment [28,42,43]. Recently, Reynoso et al. [31], suggested the use of an MG parameter that detects not only the amount of damage, but also the most frequent type of genetic damage (most frequent migration group). In the case of G, the MG with the highest frequency and with the greatest migration corresponds to the highest concentrations. A similar situation was observed for FAENA and TACKLE, while in CENTELLA, the migration range decreased but the frequency of the migration groups increases. This parameter suggests that high concentrations of FAENA and TACKLE cause specific genetic damage (the most frequent migration group) [31] associated with different substances; in this case, the presence of different adjuvants is indicated. The high frequency (92%) of MG with little migration observed in CENTELLA also suggests the presence of other adjuvants.

## 4. Materials and Methods

### 4.1. Reagents Used

G (N-(phosphonomethyl) glycine)) (CAS 1071-83-6) and ethyl-methanesulfonate (EMS)( CAS 66-27-3) were purchased from Sigma Chemical Co. (Guadalajara, Jalisco, Mexico). Both dimethyl sulfoxide (DMSO) (CAS 67-68-5) and disodium salt EDTA (CAS 60-00-4) were obtained from J.T. Baker (Ciudad de México, Mexico). FCG: FAENA, TAKLE, and CENTELLA were purchased from a commercial establishment in Guadalajara. According to the fact sheet, the glyphosate content in the GBH was: FAENA (360 g/L), TACKLE (360 g/L), and CENTELLA (360 g/L), which corresponds to 74.7% monoammonium salt of N-(phosphonomethyl) glycine and 25.3% adjuvants and inert substances (see additional material).

### 4.2. Obtaining Blood Samples

Human whole blood samples, frequently used in the comet test, were obtained according to what was reported [31]. Prior informed and signed consent (see additional material) and 200 µL of whole blood were obtained via ring puncture in forty individuals (eight for each experimental group) of a legal age who were non-smokers or exposed to environmental chemical contamination or other factors that affect DNA integrity (application of a questionnaire). An amount of 100 µL was used to assess the cell viability of peripheral blood lymphocytes using the Trypan Bluey test. The mean percentage for each group was >85%. Individuals exposed to factors that represent a possible genotoxic risk were excluded. An amount of 100 µL of each sample was centrifuged at 3000 rpm for 3 min in 5 mL of phosphate buffer (PB) (160 mM NaCl, 8 mM Na_2_HPO_4_, 4 mM NaH_2_PO_4_ and 50 mM EDTA, pH 7 and 4 °C) and re-suspended again in 1 mL of PB at 4 °C until its further use. A similar process was used for the evaluation of the controls, with four minutes for the PC (EMS 10 mM) and four minutes for NC (only suspended in PB), and both positive and negative control groups were used for monitoring, as has been recommended [44].

### 4.3. Preparation of G and GBH Concentrations

The cells were exposed to G at final concentrations of 0.1, 1, 10, and 50 mM, and to three GBH at the same glyphosate-equivalent final concentrations according to information on the use sheet of each one. The maximum possible concentration of G (50 mM) was prepared and lowered to the concentrations used by Khan et al. [7]. The solutions of G and each GBH were previously prepared at twice the indicated concentrations. PB (160 mM NaCl, 8 mM Na_2_HPO_4_, 4 mM NaH_2_PO_4_, and 50 mM EDTA, pH 7) was also prepared at twice its usual concentration. The mentioned final concentrations were obtained by mixing 2 mL of the G or GBH solution to be tested with 2 mL of PB and by adding 100 μL of the previously obtained blood sample. The same procedure was used for the EMS of 10 mM. After homogenization, the tubes with the experimental solutions were kept at 4 °C for two hours and in the absence of light. Afterwards, it was centrifuged at 3 rpm for 3 min to remove the supernatant, and the process was repeated twice with 5 mL of PB. Finally, the pellet was suspended in 1 mL of PB. These blood cells were subsequently placed on agarose gels. In the case of the negative control, the sample previously obtained in the PB was used. This procedure was performed twice for each sample of the eight individuals. A general description of the research process is presented in Figure 4. An amount of 200 µL of whole blood per individual was collected; 100 µL were used for cell viability testing and 100 µL were suspended in 900 µL of phosphate solution. In total, 100 µL of this mixture were added to each previously prepared concentration of G or GBF. A similar process was carried out with PC and NC. Eight individuals participated in each experimental group.

### 4.4. Comet Assay

The alkaline comet assay was carried out using the method of Speit and Hartmann [45], as was mentioned by Alvarez-Moya et al. [34]. It is as follows: slides were covered with Normal Melting Point (NMP) agarose at 1%, then the agarose was allowed to solidify, and then it was removed from the slides, giving a completely clean surface as a result. Next, a 0.6% Low Melting Point (LMP) agarose layer was placed on each slide. Once it solidified, another agarose layer was added (10 µL of the suspension containing the whole blood and 90 µL of the 0.5% LMP agarose). Finally, a third layer of 0.5% LMP agarose was added to cover the second layer. After that, the slides were immersed in lysis solution (2.5 mM NaCl, 10 mM Na_2_EDTA, 10 mM Tris-HCl, 1% Sodium lauroyl sarcosinate, 1% Triton X-100, and 10% DMSO, pH 10) for 2 h at 11 °C. After this, the slides were placed in a horizontal electrophoresis system with the electrophoresis buffer (300 mM NaOH, 1 mM Na_2_EDTA, pH 13) for 45 min. Electrophoresis was then carried out for 30 min at 1.0 V/cm with an amperage of ~300 mA and between 10–15 °C. Thereafter, the slides were washed with distilled water and stained with 90 μL of ethidium bromide (100 μL at 20 μg/mL) for 3 min. Immediately after, the slides were immersed in distilled water for three minutes. Finally, the slides were rewashed with distilled water for 15 min.

### 4.5. Observation and Quantification of Comets

A fluorescence microscope with an excitation filter of 515–560 nm was used for the quantification of the comets. The tail length was measured with the Comet Assay electrophoresis System II software (4250-050-ES) (ZEEIZ SINOPTIC MIKRO S.A DE C.V, Guadalajara, Mexico, 2012). Approximately 200 cometized cells per studied subject were analyzed [31].

### 4.6. Statistical Analysis

The statistical software StatPlus 2 was used to perform an analysis of variance (ANOVA), a Fisher test, and an F-Test for the variances of two samples. A confidence level of 0.05 was used.

## 5. Conclusions

We show that pure glyphosate and commercial GBH (Faena, Tackle and Centella formulations) give signals of genotoxicity in human blood samples in the comet assay. The genotoxicity increased in the formulations, indicating genotoxic activity also in the added adjuvants present in these products. The use of the MG parameter allowed us to detect a certain type of genetic damage associated with different formulations.

## Figures and Tables

**Figure 1 ijms-24-04560-f001:**
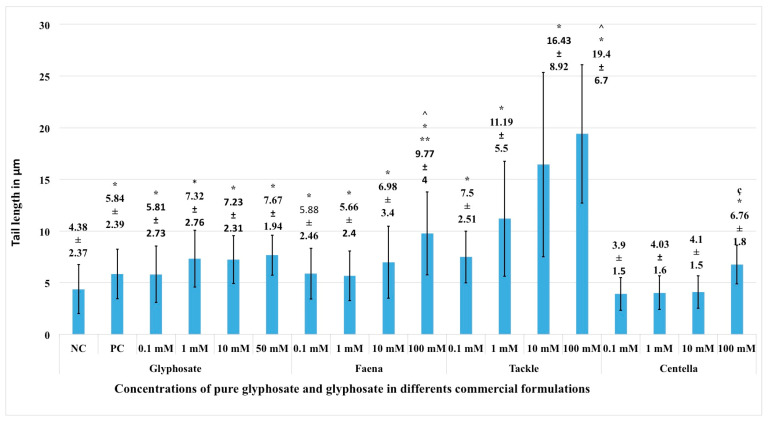
Tail length parameter. Genetic damage in blood cells exposed to different concentrations of G and in three glyphosate-based herbicides GBH. Note: G in each GBH was calculated according to informative paper, so the evaluated concentrations of G were similar to G. NC, negative control (phosphate buffer), PC, positive control (10 mM EMS). * Statistical difference with respect to NC (*p* < 0.05). ^ Statistical difference of G 50 mM with respect to FAENA 100 mM and TACKLE 100 mM (*p* < 0.05). ** Statistical difference between FAENA 100 mM and TACKLE 100 mM (*p* < 0.05). ç Statistical difference of CENTELLA 100 mM with respect to FAENA 100 mM and TACKLE 100 mM (*p* < 0.05).

**Figure 2 ijms-24-04560-f002:**
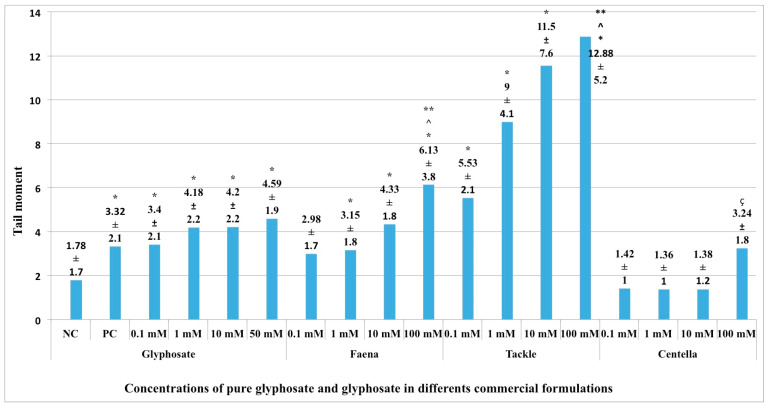
Tail moment parameter. Genetic damage in blood cells exposed to different concentrations of G and in three commercial formulations. Note: G in each CFG was calculated according to informative paper, so the evaluated concentrations of G were similar to G. NC, negative control (phosphate buffer), PC, positive control (10 mM EMS). * Statistical difference with respect to NC (*p* < 0.05). ^ Statistical difference of G 50 mM with respect to FAENA 100 mM and TACKLE 100 mM (*p* < 0.05). ** Statistical difference between FAENA 100 mM and TACKLE 100 mM (*p* < 0.05). ç Statistical difference of CENTELLA 100 mM with respect to FAENA 100 mM and TACKLE 100 mM (*p* < 0.05).

**Figure 3 ijms-24-04560-f003:**
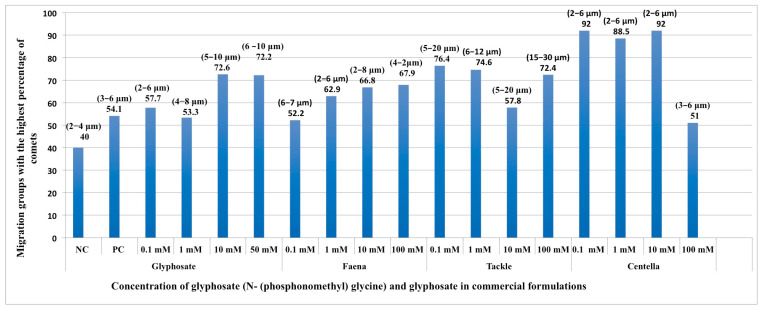
Migration groups parameter. Genetic damage in blood cells exposed to different concentrations of G and in three commercial formulations; the most frequent migration group observed for each evaluated substance is shown. NC, negative control (phosphate buffer), PC, positive control (10 mM EMS).

**Figure 4 ijms-24-04560-f004:**
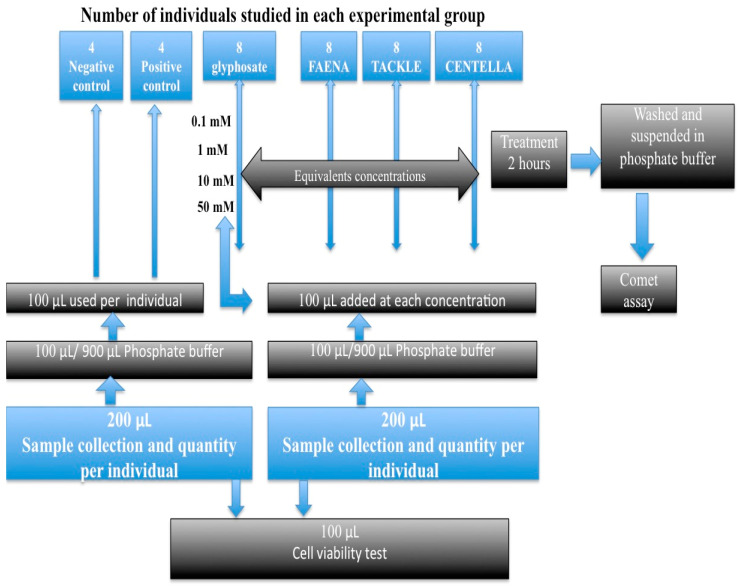
Diagram of the research process beginning with the sample collection.

## Data Availability

The data presented in this study are available on request from the corresponding author.

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
