# Peer review of "Assessment of Genetic Damage Induced via Glyphosate and Three Commercial Formulations with Adjuvants in Human Blood Cells"

_ijms, 2023, doi:10.3390/ijms24054560_

Round 1

Reviewer 1 Report

Review of Alvarez-Moya and Reynoso-Silva, “Assessment genetic damage induced for glyphosate….”

General comments:

This paper takes up a very interesting and important topic that is under-research. I therefore recommend it for publishing. However, the manuscript must be improved language wise. It contains too many weak sentences and have a large potential for improvements. The authors should also search for relevant effect of glyphosate / GBHs studies in animals.

Specific comments: suggestions in quotes, [my comments in these parentheses]

Title should be shortened, a suggestion could be: “Assessment of genetic damage induced by glyphosate and three commercial formulations with adjuvants in human blood cells”

Line 14. “……. and three commercial glyphosate-based herbicides (GBH)” [I recommend the use of GBH as the short version of these products, rather than CFG (in the whole paper), since I have seen numerous articles that used GBH.]

Line 15: “Methods: Human blood cells were….”

Line 17: “..,was observed in all concentrations with pure glyphosate and with FAENA and TACKLE formulations. These two…”

Line 19: “….in a higher proportion compared to pure glyphosate only.”

Line 32: “…. as commercial glyphosate-based herbicide (GBH) formulations (refs), which have additional compounds (adjuvants)….”

Line 36: [There exists more literature on this, e.g. Glyphosate and aminomethylphosphonic acid (ampa) in urine of children and adolescents in germany–human biomonitoring results of the german …

N Lemke, A Murawski, MIH Schmied-Tobies… - Environment …, 2021 - Elsevier

Glyphosate and AMPA in human urine of HBM4EU-aligned studies: part B adults

J Buekers, S Remy, J Bessems, E Govarts, L Rambaud… - Toxics, 2022 - mdpi.com

… of urine, divided by the body weight multiplied by urinary excretion … is the urinary excretion
fraction. It is the ratio of the mass of glyphosate excreted in urine over the mass of glyphosate

Lagre Referanse Sitert av 4 Alle 12 versjoner

[PDF] mdpi.com

Glyphosate and AMPA in Human Urine of HBM4EU Aligned Studies: Part A Children

J Buekers, S Remy, J Bessems, E Govarts, L Rambaud… - Toxics, 2022 - mdpi.com

… on the exposure of children to glyphosate (Gly) in Europe. … Median Gly concentrations in
urine were below or around … salivary gland of rats) indicated no human health risks for Gly in the …

Lagre Referanse Sitert av 3 Alle 9 versjoner

[HTML] sciencedirect.com

[HTML] Characterization of glyphosate and AMPA concentrations in the urine of Australian and New Zealand populations

G Campbell, A Mannetje, S Keer, G Eaglesham… - Science of the Total …, 2022 – Elsevier

Some of these may be referenced]

Line 41: [Should it not be IARC rather than AIRC??]

Line 45-46: “… considered confidential business information (CBI), there is no…”

Line 99: “… were equal to G in the GBHs

Line 108: “…  concentrations showed significant genotoxic activity” [Meaning gets wrong in the text here!]

Line  148: [describe what negative and positive controls are under the figure]

Figure 3. [It seems to me that Centella (and Tackle?) at very high concentration gives an antagonistic response, which could be mentioned, e.g. in lines 168-173]

Line 151. “… is evidence of genotoxic….”

Line 185. “…. with the highest frequency..”

Line 187: “… A similar situation…”

Line 188. “… range decreased but the frequency….”

Line 191. “….. (92%) of MG….”

Line 201. “…… According to the fact sheet…” [or material data sheet?]

Line 202. [I don’t understand that there is three formulations but only two percentages (74.7 and 25.3%). Please clarify paragraph]

Line 264. “We show that pure glyphosate and commercial glyphosate-based herbicides (Faena, Tackle and Centella formulations) give signals of genotoxicity in human blood samples in the comet assay. The genotoxicity increased in (two of?) the formulations, indicating genotoxic activity also in the added adjuvants present in these products.” [the present conclusion is too short and can be extended]

Author Response

Reply to reviewer 1 attached to document

Reviewer 2 Report

Dear authors.

The use of glyphosate is controversial. There are repeated reports of its hazardous properties. However, this is unfortunately not always properly documented. Of particular concern is the use of this compound to dry out plants immediately prior to seed harvesting. 

Therefore, all studies are an important input to assess the safety of its use and risks to humans. In your study, you chose a very good method, but one that was difficult for some researchers. Unfortunately, the methodology presented is not very precise and the lack of photographic documentation showing images of comets is questionable.

In addition, it would be useful to include a diagram of the research process, preferably in the form of a figure.

I have the following comments on the paper. 

1. Why were such high concentrations chosen for the study, is there a possibility of exposure to the preparation in the concentrations tested?

2. Whether cytotoxicity was assessed for the cells tested. The observed effects could have been due to apoptosis?

3. Please add the deviation on the graph in classical form, adding them as numbers is not very readable.

4. What cells were isolated, lymphocyte cultures are used. Method of isolation, culture. 

5. How many comets were analysed, no example photos of comets with preparations in different concentrations. Absence does not allow assessment of the quality of the research. The images in the supplement cannot be read.

6. Conclusions unfortunately need to be expanded. 

Author Response

Reply to reviewer 2 attached to document

Round 2

Reviewer 2 Report

Dear authors. 

Many thanks for your answers to the questions and the corrections made to the text. The work presented is interesting and addresses an important issue.